# Crosstalk between Mycotoxins and Intestinal Microbiota and the Alleviation Approach via Microorganisms

**DOI:** 10.3390/toxins14120859

**Published:** 2022-12-06

**Authors:** Daiyang Xia, Qianyuan Mo, Lin Yang, Wence Wang

**Affiliations:** 1Guangdong Provincial Key Laboratory of Animal Nutrition and Regulation, College of Animal Science, South China Agricultural University, Guangzhou 510642, China; 2Guangdong Laboratory for Lingnan Modern Agriculture, Guangzhou 510642, China

**Keywords:** mycotoxins, gut microbiota, cross-talk, alleviation

## Abstract

Mycotoxins are secondary metabolites produced by fungus. Due to their widespread distribution, difficulty in removal, and complicated subsequent harmful by-products, mycotoxins pose a threat to the health of humans and animals worldwide. Increasing studies in recent years have highlighted the impact of mycotoxins on the gut microbiota. Numerous researchers have sought to illustrate novel toxicological mechanisms of mycotoxins by examining alterations in the gut microbiota caused by mycotoxins. However, few efficient techniques have been found to ameliorate the toxicity of mycotoxins via microbial pathways in terms of animal husbandry, human health management, and the prognosis of mycotoxin poisoning. This review seeks to examine the crosstalk between five typical mycotoxins and gut microbes, summarize the functions of mycotoxins-induced alterations in gut microbes in toxicological processes and investigate the application prospects of microbes in mycotoxins prevention and therapy from a variety of perspectives. The work is intended to provide support for future research on the interaction between mycotoxins and gut microbes, and to advance the technology for preventing and controlling mycotoxins.

## 1. Introduction

Mycotoxin is a naturally occurring substance produced by fungi. Consumption of low concentrations of mycotoxins in animals would result in severe hazardous symptoms [1]. The first credible evidence of the hazardous of mycotoxins effects dates back to the 11th century when ergot intoxication caused widespread human and animal poisoning and even mortality in Europe [2]. With detection and analysis technology advances, hundreds of mycotoxins have been discovered [3]. Five of these mycotoxins, aflatoxin B1 (AFB1), deoxynivalenol (DON), zearalenone (ZEA), fumonisin B1 (FB1), and ochratoxin A (OTA), have historically been major objects of mycotoxin study because of their high detection rates and significant toxicities in feed raw materials and foodstuffs [4]. Due to the extensive prevalence of fungi in the environment, grains have been contaminated with mycotoxins during the growth process, and almost all agricultural commodities are susceptible to fungi infection and the production of mycotoxins if improper storage [5]. Mycotoxins not only lower livestock productivity and result in significant economic losses, but also pose a threat to human health because of their accumulation along the food chain. Severe clinical symptoms occur during animal breeding with mycotoxin-contaminated feed, including diarrhea, liver and kidney damage, pulmonary edema, vomiting, bleeding, and tumors [6,7,8,9,10,11]. Additionally, mycotoxins have a synergistic effect, typically including a combination of toxins, which results in human and animal poisoning upon consumption [12]. This complicates the toxicological mechanism of mycotoxins.

The gut serves as the first line of defense and protection against mycotoxins and the first location of mycotoxins’ absorption into the body [13,14]. The gut microbiota plays a critical role in forming the intestinal barrier and maintaining intestinal homeostasis [15]. With the introduction of new concepts such as the brain-gut axis, liver-gut axis, and kidney-gut axis [16,17,18,19], as well as the widespread use and advancement of microbial sequencing technology, an increasing number of studies are focusing on the impact of mycotoxins in intestinal microbiota. Meanwhile, scientists are investigating the role of intestinal microbial changes in the process of mycotoxin poisoning and detoxification. The deleterious effects conferred by mycotoxins-induced microbial alterations would be brought to the forefront for investigating toxicological effects and response of the host.

Interpreting the crosstalk between various mycotoxins and microbes will be crucial for mycotoxin control and prevention. The crosstalk mechanisms between AFB1, DON, ZEA, FB1 and OTA with intestinal microbes are described in this review. The potential for microbial applications in mycotoxin hazardous mitigation is also discussed with present viewpoints. This paper aims to shed light on the interaction between various mycotoxins and microbes, discover new toxicological processes for mycotoxins, and identify potential treatment targets.

## 2. Toxicity of Mycotoxins to Intestinal Epithelial Cells

As the “transit point” of animals ingesting mycotoxins, intestinal epithelial cells are the first barrier where mycotoxins contact the animal body [14]. Numerous animal studies have demonstrated that mycotoxins such as DON, ZEA, FB, OTA and AFB1 trigger direct epithelial cell damage (Table 1). The most noticeable symptom is that mycotoxin directly limits the growth and structural destruction of the small intestinal villi, the outer wall consisting of a single layer of epithelial cells [20,21,22,23]. Recently, studies on the mechanism of directive injury of mycotoxins on the intestinal epithelium were well investigated. By studying changes in the physiological functions of intestinal cells exposed to mycotoxins, researchers initially explored the direct toxicological effects of mycotoxins in the absence of gut microbiota and have achieved considerable progress.

### 2.1. DON

The disclosure of the direct toxicological effects of DON on the intestinal epithelium has made great progress, and a systematic explanation mechanism has been constructed from multiple perspectives. It is assumed that the detrimental effects of DON on intestinal epithelial cells are mediated predominantly through the following three pathways: (Ι) DON activates DAO, up-regulates NF-κB signaling pathway, increases the levels of inflammatory cytokines in the intestine, and ultimately mediates intestinal epithelial apoptotic [24,25,26]. (II) DON decreases the expression of the trefoil factor family (TFFs) of peptides (a type of bioactive substance that could govern tissue regeneration, improve barrier function, and decrease proinflammatory expression) via triggering the MAPK signaling pathway, hence inhibiting intestinal epithelial cell self-repair. This mechanism has been validated in human intestinal cell line HT29-16E and swine intestinal explants [24,27]. (III) Recently, a new perspective on the enterotoxicity of DON has also been proposed. It was revealed that DON can also decrease the stability of intercellular compact proteins in the intestinal epithelium [28]. Along with accelerating tight junction protein degradation in the fusion medium, DON would activate the p38 (MAPK) signaling pathway, resulting in the swallowing and degradation of Occluding and ZO-1 in lysozyme, and eventually destroying the small intestine villus structure and increasing intestinal permeability. Collectively, DON can activate immune pathways and induce inflammatory responses in the intestinal epithelium. And it can activate lysosomes to engulf connexins between intestinal epithelial cells, resulting in structural collapse. In addition, DON also inhibits the self-repair process of the intestine, which eventually leads to the death and autophagy of intestinal epithelial cells.

### 2.2. ZEA

ZEA generates oxidative stress at the cellular level and increases lactate dehydrogenase activity, further impairing the organism’s scavenging of reactive oxygen species (ROS) and increasing the amount of oxidative stress in intestinal cells [29,30]. In addition, ZEA also mediates the activation of NLRP3 inflammasome in mouse intestine, which in turn promotes the level of Caspase-1, up-regulates inflammatory cytokines, leads to the expansion of inflammatory cells in intestinal epithelial cells, and produces apoptosis [31]. On the other hand, ZEA also leads to aberrant G2/M transition in IPEC-J2 cells by disrupting the cell cycle signaling system, consequently reducing cell proliferation and producing intestinal epithelial damage [32]. Intestinal epithelial cells have the shortest cell cycle and are the fastest-renewing somatic cells. Therefore, the perturbation of cell cycle signals by ZEA will greatly inhibit the development and self-repair process of the intestinal epithelium.

### 2.3. OTA

The mechanism of OTA cytotoxicity in the intestinal epithelium is mostly based on the production of ROS and the stimulation of apoptosis-regulating genes. Wang et al. [33] found that OTA may generate reactive oxygen species (ROS) in IPEC-J2 cells, which elevated the activity of the Ca^2+^ and MLCK Signaling pathways and ultimately resulted in barrier malfunction and destruction. Comparative transcriptomics demonstrated that OTA enhanced the expression of apoptosis-related genes such as *casp3*, *cdc25B* and *egr1* in Caco-2 cells, elucidated the genome-wide biological reaction perspective of OTA controlling intestinal epithelial damage [34]. The researchers also explained the toxicological mechanism of OTA on intestinal epithelial cells from various aspects. It is worth noting that OTA appears to be a dose-dependent amplifier of apoptotic signaling to intestinal exposed-epithelial cells. OTA showed a perturbation of functional gene expression of human intestinal cells at a very low dose (0.0005 μg/mL). 

### 2.4. FB1

FB1 regulates the secretion of mucin by stimulating the ERK phosphorylation pathway, lowering the expression level of intestinal tight junction protein, and inhibiting the viability of IPEC-J2 cells [35]. Mucin secreted by intestinal epithelial goblet cells is a glycoprotein composed of mucopolysaccharides that protects intestinal cells. The network of mucins makes it difficult for chemical irritants, digested foods, toxins, and bacteria to pass through, protects the intestinal epithelium from damage, and prevents pathogens from binding to the intestinal epithelium. The digestion of mucin by FB1 is likely to be an important inducement for it to enter the body and cause multi-organ toxicity. It is worth noting that FB1 could regulate the mouse intestinal aryl hydrocarbon receptor (AHR), the constitutive androstane receptor (CAR), the pregnane X receptor (PXR), and downstream target genes (CYP450s) to disrupt nuclear xenobiotic receptor (NXR) homeostasis, and meanwhile induce intestinal villus and epithelial layer shedding, intestinal gland atrophy, and necrosis [36].

### 2.5. AFB1

There are few studies on the mechanism of AFB1 intestinal epithelial toxicity. By observing the ultrastructure of intestinal epithelial cells exposed to AFB1, the researchers found that AFB1 is significantly toxic to the organelles of intestinal epithelial cells. AFB1 causes mitochondrial vacuolization in small intestinal cells, leads to the disappearance of mitochondrial cristae, junctional complexes, and terminal reticulum, and then induces apoptosis [37]. Notably, AFB1 also reduced the ratio of goblet cells in epithelial cells in this study. This result indicates that FB1 and AFB1 also seem to have certain effects on the differentiation of intestinal epithelial cells, but whether the underlying mechanism is the toxic effect on goblet cells or the direct induction of the cell differentiation process remains to be further investigated. In addition, researchers have revealed that AFBI exposure increases p42/44 (MAPK) phosphorylation in Caco-2 cells, inhibiting tight junction protein synthesis between epithelial cells, increasing intestinal permeability, and weakening the intestinal barrier [38].

In conclusion, based on the results of direct mycotoxin exposure assays on intestinal epithelial cells, mycotoxins usually directly inhibit the normal division and proliferation of intestinal epithelial cells and cause apoptosis through modulation of cell signaling pathways.

## 3. Crosstalk between Mycotoxins and Intestinal Microbiota

Regarding the critical function of the gut microbiota in maintaining gut homeostasis, the enterotoxicity of mycotoxins in animals is inseparable from the interaction with the intestinal microbiota. In recent years, increasing experiments have focused on gut microbiota changes under the stress of mycotoxin exposure (Table 2). Mycotoxins have been proven to disrupt intestinal microbiota community homeostasis, and the underlying mechanisms have been gradually explored.

### 3.1. DON

It is now widely recognized that DON and OTA can exacerbate harmful effects by altering intestinal bacteria. Exposure to DON at either long-term low-dose or short-term medium-to-high doses significantly perturbed the gut microbial composition of animals. When using pigs as experimental animal models, DON can directly affect the variety of gut microbiota. In the short term, the composition of fecal bacteria may be drastically altered [39,40]. In addition to perturbing the gut microbial community as a whole, DON operates synergistically with pathogenic bacteria, aggravating the disease process in the host. Ruhnau et al. [41] proved that DON collaborated with *Campylobacter jejuni* to enhance intestinal load, disrupt the intestinal barrier, and expedite intestinal pathogen translocation into the liver and kidney. In addition, DON is also demonstrated to enhance the relative abundance of *Clostridium perfringens* in the jejunum of broilers, thereby increasing the digestive load [42]. Meanwhile, DON enhanced the abundance of pathogenic *Escherichia coli* in the colon and exacerbated the Escherichia-induced intestinal cell DNA damage [43,44]. Metagenomic analysis revealed that DON severely disrupted the intestinal microbiota in mice and impaired biosynthesis and repair functions. A high dose of DON damages the nerve sphingolipid, protein digestion and absorption, and lipoic acid metabolism pathways [45].

### 3.2. OTA

Regarding the perturbing effect of OTA on gut microbes, many studies have shown that OTA had a substantial effect on the relative number of *Firmicutes*, increased the relative abundance of *Bacteroides*, and decreased the β diversity of the intestinal microbiota [46]. Notably, Wang et al. [47] explored the OTA-induced alterations via intestinal microbiota in its toxicological mechanism and estimated that OTA-mediated liver inflammation was microbially dependent. OTA increased the relative abundance of lipopolysaccharide-producing bacteria (*Bacteroidetes*), elevated the intestinal LPS load, weakened the intestinal barrier, and ultimately induced liver inflammation via LPS-specific activation of the liver TLR4/MyD88 signaling pathway. The researchers constructed a pseudo-sterile animal model with antibiotics, and the fecal bacteria transplantation experiment proved that OTA-induced changes in gut microbes are an important incentive for OTA-induced liver validation.

### 3.3. ZEA

When animals were exposed to ZEA alone, no substantial changes in their gut bacteria occurred, and ZEA had no discernible influence on the composition of intestinal microbes. By stimulating the RhoA/ROCK pathway, ZEA tended to inhibit microbial involvement in intestinal mucin formation [48,49,50]. These results suggest that ZEA may have little effect on the overall microbial community in the gut, but through its effect on the mucus layer of the intestinal epithelium, it is likely to have some effects on the parasitic bacteria that adhere to mucin or decompose mucin. This series of changes may further negatively affect the construction of the intestinal barrier. Studies have shown that ZEA has a significant impact on the overall functional gene level of gut microbes. ZEA inhibits intestinal microorganism glycerophospholipid metabolism, which may be one of the reasons contributing to ZEA-induced ovarian reproductive damage [51,52].

The combined presence of more than two mycotoxins frequently occurs in naturally moldy cereals and grains. Therefore, it is more applicable for investigation in combined mycotoxins impact on gut microbiota. As ZEA combined with DON, the organization of the intestinal microbiota was drastically altered [53]. High doses of DON and ZEA changed the number of proteins involved in microbial metabolism, genetic processing, and oxidative stress responses (related to the ribosome and pentose phosphate pathways), resulting in structural modification of the gut microbiome [54]. Additionally, combined ZEA and DON suppressed costimulatory molecule expression on intestinal CD^4+^ T cells and il-4R-mediated Th2 cell development, hence impairing intestinal resistance and pathogen clearance [56]. These results reflected the additive effect among mycotoxins, whereas, in reality, multiple mycotoxins are often combined and generate in grains and feeds. ZEA has a high detection rate in grains, therefore the combined effects of other toxins and ZEA, and the harmful effects of other mixed mycotoxins on intestinal microbes are urgently needed to advance research.

### 3.4. AFB1

AFB1 exposure also induces gut microbial dysfunction. Fermentation of carbohydrates and proteins and amino acids to produce short-chain fatty acids (SCFA) is an important function of intestinal microorganisms. SCFA produced by intestinal microorganisms is not only an important source of energy for intestinal cells but also drives the absorption of nutrients and hormone production in the intestine. After four weeks of AFB1-exposed, SCFA, pyruvate-related pathways, amino acids, bile acids, and long-chain fatty acid metabolic pathways were all eliminated in the intestinal bacteria of rats. The efficiency of the host digestive system, energy supply, intestinal immunity, neurotransmitter synthesis, and enterohepatic crosstalk was also affected [55]. When studying the hazards of exposure to mycotoxins, researchers tend to overlook this indirect effect of gut microbial dysbiosis. Little is known about changes in gut microbial communities under mycotoxin exposure. As an important “organ” that performs host digestion, metabolism, absorption, and immune processes, the gut microbiome deserves more attention. There is little experimental evidence on the changes of intestinal microorganisms in animals exposed to FB1, therefore we do not summarize FB1 related information in this part. With the increasing amount of research focusing on changes in the gut microbiome upon exposure to mycotoxins, the understanding of the role of the gut microbial community under toxin perturbation will become improved. This will also contribute to a more complete revelation of the underlying toxicological mechanisms of mycotoxins.

## 4. Alleviate Mycotoxins Harm by Microbiota

As the crosstalk between mycotoxins and animal gut microbiota discussed, mycotoxin-induced damage in the host organism is partially attributed to microbiota-mediated enterotoxicity and pathological changes. Thus, strengthening the intestinal barrier by modulation of the gut microbiota may represent a unique strategy for minimizing the harm caused by mycotoxins.

At the present, there are three primary strategies for mitigating the harm caused by mycotoxins transmitted by microorganisms: (Ι) directly degrade mycotoxins in food through microbial pretreatment and reduce toxin intake; (II) by increasing probiotic colonization, the formation of bacterial-toxin complexes is inhibited, thereby reducing toxin absorption in the intestinal tract; and (III) by regulating the intestinal microecology with probiotics or prebiotics, remodeling the intestinal microflora, enhancing the intestinal barrier, relieving intestinal toxicity, and reducing toxin penetration. 

As several recent pieces of literature reviewed the mycotoxin degradation by microorganisms, we summarized the topic of mycotoxin degradation by microbiota in Section 4.1. This review focused here on the alleviation of mycotoxins in the host by reshaping and modulating intestinal microorganisms via microbiota adhesion effect or supplemental nutrients etc. 

### 4.1. Alleviate Mycotoxins Harm by Microbiota Degradation

With the development and utilization of microbial culturomics and sequencing technology, an increasing number of microorganisms may be cultured individually in the laboratory. Recently, an increasing number of microorganisms have been shown to degrade mycotoxins with high efficiency and specificity. 

We list the microbes that can directly and efficiently degrade mycotoxins (Table 3). With the advancement of food fermentation technology and the pre-storage treatment process, it is anticipated that the direct elimination of toxins by microorganisms will become increasingly prevalent.

While microbial degradation of mycotoxins has the advantage of being highly efficient and specific, it also has drawbacks, including limited degradation efficiency of mycotoxins complexes, severe pretreatment conditions, and complex fermentation products.

### 4.2. Alleviate Mycotoxins Harm by Microbiota Adhesion Effect

Mycotoxins in feed and food are typically in the form of mycotoxins-glycoconjugates generated from plants. These conjugates break down hardly in vivo and are often not digested in the stomach or small intestine, resulting in large intestinal colon toxicity. Increased colonization of high-adhesion probiotics results in the formation of bacteria-toxin complexes in the intestinal segment, which effectively reduces toxin absorption in the large intestine [73]. Lactic acid bacteria adsorption capacity for mycotoxins is a prominent study area in recent studies, owing to their distinctive secretory mucin layer with strong adhesion and their ease of cultivation [74]. 

The adsorption capacity of ZEA by lactic acid bacteria was first reported in 2017 and examined the durability of the ZEA-lactic acid bacteria combination. It was found that ZEA had a 68.2 percent adsorption capacity on lactic acid bacteria and that 15.8 percent of ZEA remained adsorbed in the complex particles after three bleaching cycles. The potential of lactic acid bacteria as ZEA bioactive adsorbents was demonstrated in this study [75]. With the advancement of microbial culturomics, increasing isolated *Lactobacillus* species were assessed for the adsorption impact, and the FTIR (Fourier Transform Infrared) technique was utilized to determine the mechanism of microbial adsorption. *Lactobacillus plantarum* has been found to limit the bioavailability of ZEA in the gastrointestinal system by adhesion, hence reducing the genotoxicity and nephrotoxicity of ZEA [76].

ZEA was absorbed by *L. plantarum* BCC 47723 with the effect of hydrophobic adsorption rather than electrostatic adsorption. The adsorption efficacy of *L. plantarum* could be increased further by altering the bacterial cell structure via heat treatment [77]. 

High-performance liquid chromatography analysis indicated that *Lactobacillus paracei* might decrease ZEA to α-Zol and β-Zol, resulting in less harmful byproducts [78]. Król et al., found that *Lactococcus lactis* is capable of neutralizing ZEA via bacterial proteins and deprotonated carboxyl groups in peptidoglycans (asparagine and glutamine) [79]. 

Except for ZEA, lactic acid bacteria were also capable of adsorbing AFB1 and DON [79,80]. It is demonstrated that *Lactobacillus Plantarum* could adsorb 82% of AFB1 in vitro and that the resulting complex retained high stability after five washes [81]. It was also reported that *Lactobacillus para cel* live cells bind AFB1 98 percent of the time and dramatically lower AFB1 levels in the serum of rats exposed to the toxin [82]. *Lactobacillus para cel* LHZ-1, isolated from yogurt, the cell wall had the highest rate of DON absorption, up to 40.7 percent, supporting that the S protein released by the *lactobacillus* cell wall had a major role in adhesion [83]. 

### 4.3. Alleviate Mycotoxins Harm by Shaping the Microbiota

As shown in the table below, many studies have demonstrated the detoxification effect of additional supplemental nutrients such as non-flavonoid polyphenols, flavonoids, dietary fibers, terpenoid carotenoids, and fatty acids on the toxic effects of mycotoxins (Table 4); the crosstalk between the aforementioned substances and gut microbes, however, was not considered. The gut microbiota is an important link that maintains the host’s metabolic function, immune function, and nutrient absorption. Recently, growing studies have revealed the effects of nutrients on gut microbes and entire metabolic processes passing through the digestive tract. Combining the potential mechanisms of nutrients to mitigate mycotoxins hazards, as well as the interactions between nutrients and gut microbes, will help us to clarify the possibility of gut microbes as potential targets for antagonizing mycotoxins.

#### 4.3.1. Non-Flavonoid Phenolic

Polyphenolic compounds (such as curcumin, caffeic acid, resveratrol, etc.) have an intense alleviation effect on damages caused by mycotoxins.

Curcumin is known as a polyphenolic compound with pharmacological activities, such as antioxidative, anti-inflammatory, and antibacterial. Previous studies revealed that curcumin supplementation attenuated liver damage induced by AFB1 exposure in chickens, mice, and rats and reduced liver inflammation in OTA-exposed ducks [84,85,86,87,88]. Increasing pieces of evidence demonstrated that the hepatotoxicity of AFB1 and OTA is microbe-dependent. AFB1 and OTA disrupt animal intestinal microbial homeostasis, increase the relative abundance of lipopolysaccharide-producing *Bacteroides*, and ultimately induce liver inflammation in animals [47,148]. As a natural herbal polyphenol, curcumin has been reported in several studies to modulate the composition of gut microbes. Curcumin could increase the relative abundance of *Lactobacillus* in the intestine of mice, and reduce the relative abundance of *Shigella Enterobacter* and pathogenic *Bacteroides*, reducing lipopolysaccharide production to alleviate the process of metabolic endotoxemia [149,150]. In addition, curcumin significantly up-regulates the relative abundance of the butyric acid-producing bacteria *Butyricicoccus* in the rat gut, and the species *Butyricicoccus* repair intestinal barrier damage and further improve liver lipids by degrading fermented proteins and carbohydrates to produce short-chain fatty acids [151].

As a powerful antioxidant, resveratrol effectively alleviates liver injury in mice exposed to AFB1 and can improve serum immune indexes in rats exposed to ZEA. Notably, multiple studies have demonstrated the protective effect of resveratrol on the gut of weaned piglets exposed to DON. Significantly, resveratrol up-regulated the relative abundance of SCFA-producing bacteria, such as *Butyricicoccus*, *Ruminococcus_1*, *Roseburia* and *Adlercreutzia* in rat gut, and increased the level of SCFA in feces [152,153]. During the weaning stage of piglets, the level of secondary bile acids in the intestine was up-regulated due to dietary changes, resulting in disturbance of the intestinal microbiota and intestinal barrier [154,155]. Long-term exposure to DON also induces bile duct hyperplasia and tissue damage in piglets [156]. Interestingly, studies have reported that resveratrol could mediate the synthesis and recycling of bile acids by affecting gut microbes. Resveratrol increases the relative abundance of *Lactobacillus* and *Bifidobacterium*, thereby increasing the activity of bile salt hydrolase. And reducing the concentration of bile acids in the ileum. Meanwhile, it has also been found to up-regulate the intestinal FXR signaling pathway and enhance the intestinal barrier [157]. Therefore, resveratrol is likely to mediate the repair effect of gut microbes on the gut of weaned piglets exposed to DON.

Caffeic acid is a hydroxycinnamic acid-related organic compound with both phenolic hydroxyl and acrylic functional groups. It possesses antioxidant, immunomodulatory, and anti-inflammatory properties. A recent study showed that caffeic acid alleviates AFB1-induced kidney damage in rats. Dietary supplementation of caffeic acid significantly reduced the relative abundance of *Bacteroidetes* and *Turicibacter* and increased the abundance of *Alistipes*, *Dubosiella* and *Akkermansia* [158,159]. *Alistipes* were shown to produce SCFA, and *Akkermansia* is a novel probiotic that could effectively reduce the potential risk of colitis by regulating the secretion of mucin [160,161]. Therefore, gut microbiota may be potential targets for caffeic acid to exert its detoxification effect.

#### 4.3.2. Flaudio-Videoonoids

Flaudio-videoonoids substances (e.g., anthocyanin, gallic acid, quercetin and baicalin and their derivatives) are also shown detoxification effects against mycotoxins.

Anthocyanin is a water-soluble plant pigment. Toxicological studies showed that anthocyanins alleviate the damage caused by AFB1 in broilers and the damage caused by ZEA in mice. As polyphenols naturally extracted from plants, anthocyanins not only play an antioxidant role, but also significantly change the composition of animals’ gut microbes. Anthocyanin supplementation reduces intestinal inflammation by increasing the relative abundance of *Lactobacillus*, *Bifidobacterium*, *Lachnospira* and *Ruminococcus*, which in turn increases SCFA production, improves intestinal barrier and mucus production, and reduces the potential risk of intestinal inflammation [162,163].

Gallic acid is not only a polyphenolic compound present in plants, but also one of the important metabolites of gut microbes [164]. Gallic acid alleviates the reproductive toxicity of ZEA-exposed mice and the toxic effect of OTA on catfish. Adding gallic acid to the diet significantly upregulates gut microbial diversity in dogs and bees [165,166]. Recent studies have shown that ZEA is likely to induce reproductive toxicity by disrupting the blood-testis barrier [167]. The protective effect of probiotics on the blood-testis barrier has gradually been discovered and clarified by researchers. *Proteobacteria* have been reported to increase the probability of semen hyperviscosity, while *Lactobacillus* improves mitochondrial activity and alleviates oxidative stress in sperm cells [168]. Interestingly, the relative abundance of probiotics such as *Lactobacillus* and *Prevotaceae* were significantly up-regulated and the level of *Proteobacteria* was significantly decreased after gallic acid was added to the diet of colitis model rats [169]. Therefore, gallic acid is likely to be able to alleviate the reproductive toxicity caused by ZEA by regulating the composition of gut microbes, and its antioxidant properties also resist the oxidative stress caused by ZEA.

In recent studies, researchers have revealed the alleviation effect of quercetin on metabolic diseases such as hyperlipidemia, hyperglycemia, and obesity [170,171,172]. Quercetin also effectively alleviates OTA-induced immune toxicity in chickens and nephrotoxicity in rats exposed to OTA. As gut microbes may be an important bridge for quercetin to regulate energy metabolism disorders, the effect of quercetin on intestinal microbes is also an inevitable consideration in the process of alleviating mycotoxins. Studies have shown that quercetin is effective against microbial dysbiosis, restores gut microbiota in antibiotics-treated mice, and significantly increases the relative abundance of butyrate-producing bacteria [173]. Similarly, a dietary quercetin supplementation in pigs experiment revealed that when quercetin was added to piglet feed, quercetin significantly upregulated *Akkermansia muciniphila* in the gut, as well as the relative abundance of SCFA-producing bacteria *Clostridium butyricum*, *Clostridium celatum* and *Prevotella copri* [174].

Baicalin is a naturally occurring glycoside that is derived from baicalein and glucuronic acid. Baicalin and its copper-zinc compounds effectively alleviate intestinal inflammation in weaned piglets exposed to DON and protect the liver and kidney toxicity in broilers exposed to ZEA. Baicalin with dietary intake is converted into baicalenin with gut microbes’ fermentation [175,176]. As baicalenin is absorbed into the host mechanism through the intestinal tract, baicalenin and baicalinit could be converted into each other under the catalysis of phase II enzymes, and phase II enzymes also have detoxification effects [177,178]. Therefore, the interaction between baicalin and gut microbes is also an important part of its detoxification effect. Dietary supplementation with baicalin monomer significantly altered gut microbial composition and induced changes in fecal bile acid profiles, according to the findings of the researchers. Moreover, baicalin inhibited the appearance of the bile acid-specific receptor FXR and increased intestinal barrier toxicity resistance [179]. This study accurately depicts the regulatory effect of baicalin on the intestinal microbiota and bile acid system and paves the way for the discovery of a more profound detoxification mechanism.

#### 4.3.3. Dietary Fiber

Many studies have shown that dietary fiber and its metabolites (oligosaccharides) also have a detoxifying effect on mycotoxins. Dietary fiber affects the peristalsis and digestion process of the intestine, and most of them are not digested in the foregut. Dietary fibers are converted into small molecular substances and absorbed by the host after the hindgut degradation and fermentation by intestinal microorganisms [180]. As the main active ingredient in moringa oleifera leaves, moringa oleifera polysaccharide significantly reduced the relative abundance of *Bacteroides* and *Helicobacter pylori* in gut microbes [122]. The protective effect of algal polysaccharides, fucoidan, and Alginate oligosaccharides on animal intestines is mainly due to their promotion of SCFA production by bacteria in the hindgut [181]. At present, the interaction effect of polysaccharides and oligosaccharides with intestinal microbes in the hindgut is still unclear. It is necessary to further design sterile or pseudo-sterile animal experiments to verify the role of intestinal microbes in the process of dietary fiber alleviating the toxicity of mycotoxins.

#### 4.3.4. Terpenoid Carotenoids

Studies have shown that two terpenoid carotenoids, lycopene, and astaxanthin, also have a significant detoxification effect on mycotoxins. Lycopene, in particular, has a relieving effect on damage with AFB1, DON and ZEA exposure. Studies show that additional dietary lycopene remodels the gut microbiome of mice with colitis, significantly reducing the relative abundance of *Proteobacteria* and increasing the relative abundance of *Bifidobacterium* and *Lactobacillus* [182]. In other studies, additional supplementation of astaxanthin in feed directly reduces the functional gene abundance that is associated with inflammation in gut microbes [183]. By regulating the abundance of functional gene fragments in intestinal microorganisms, the antagonistic effect of mycotoxin was indirectly achieved.

#### 4.3.5. Fatty Acids

Cinnamic acid and its derivatives, as well as sodium butyrate, have also shown detoxification effects on mycotoxins in a large number of animal tests. Studies in rats and chickens showed that cinnamic acid derivative ferulic acid can alleviate liver damage caused by AFB1 [98]. Laurate supplementation in the feed also shows the alleviation of broiler growth performance by the damage of FB1 [121]. A recent study in DON revealed that sodium butyrate could renovate intestinal barrier damage. SCFA plays a pivotal role in the construction of the intestinal barrier [112]. After the supply of SCFA precursors, the production of SCFA by intestinal microorganisms is also greatly promoted, and the abundance of SCFA-producing bacteria also increased. It is easier to become the dominant flora under this condition, and further, exert the biosynthetic function of SCFA.

#### 4.3.6. Microelement

Many studies have shown that mycotoxins can be detoxified well by supplementing feed with the trace element selenium. For poultry exposed to AFB1, supplementation with low doses of selenium is effective in reducing AFB1 damage to the liver and thymus of poultry [101,102]. Either selenium or yeast selenium supplementation helped to alleviate kidney damage in mice exposed to ZEA [129,135]. Yeast selenium has a strong antagonistic effect on both hepatotoxicity and nephrotoxicity in chickens exposed to OTA [145,146]. It has been demonstrated that selenium nanoparticles are effective in suppressing the relative abundance of *Enterococcus cecorum* in the intestinal tract of chickens [184]. Of note here is the selenium conjugate, yeast, which, as we mentioned earlier, is widely used for the adsorption and degradation of mycotoxins. Thus, in the process of the natural product yeast selenium acting to mitigate the toxicity of mycotoxins, yeast also plays a part in the adsorption and degradation of mycotoxins.

### 4.4. Microbiota-Guided Direct Regulatory Strategy 

In addition to intestinal toxicity, mycotoxins have complicated toxicological pathways that include hepatotoxicity, nephrotoxicity, neurotoxicity, and immunotoxicity. The concepts of the gut-liver axis, gut-brain axis, and gut-kidney axis have been proposed in recent years, prompting researchers to reconsider the role of mycotoxin-mediated intestinal microbiological disorder in extra-intestinal organ injury and pointing to a new direction for elucidating mycotoxins’ novel toxicological mechanism. It can be seen that numerous researchers have attempted to modify the gut microbiota directly through microbial community regulation.

Interestingly, certain lactic acid bacteria have a considerable protective effect against DON-induced intestinal damage. For example, *Lactobacillus rhamnosus* treatment boosts the liver antioxidant capacity, blocks the NRF2 signaling pathway, and increases butyrate synthesis by up-regulating the Buk and But genes, thereby inhibiting the IRE1/XBP1 signaling pathway and protecting mice from DON damage [185,186]. In vitro studies revealed that *Lactobacillus plantarum* supernatant was proven to significantly recover the loss of intestinal goblet cells induced by DON, and to strengthen the architecture of intestinal villi [187]. Whereas *Lactobacillus plantarum* cells have the potential to rearrange the intestinal community homeostasis disrupted by DON, and down-regulate the expression of apoptotic genes to ameliorate intestinal cell death and inflammation generated by DON [188]. Interestingly, *Lactobacillus rhamnosus* was unable to adsorb DON, but it was able to inhibit the expression of CCL_20_, IL-1β, TNFα, IL-8, IL-22 and IL-10 via MAP kinase and therefore withstand the intestinal inflammation generated by DON [189]. Notably, several probiotics were utilized to mitigate the toxicity of DON, and the effect of mending the intestinal barrier and restoring phylum-level bacteria abundance was shown in both mouse and piglet models [190,191].

The mechanism by which OTA induces liver inflammation via intestinal bacteria was detailed above, demonstrating that the LPS and TLR4 signaling pathways are critical targets of the OTA toxicological mechanism. Xia et al., revealed that melatonin successfully reversed the OTA-induced increase of *Bacteroidetes* abundance, greatly lowering LPS accumulation in the gut and liver, and relieving OTA-induced liver inflammation [192]. Additionally, other studies have demonstrated that astaxanthin and selenium-rich yeast can modulate fecal barrier function and the TLR4/MyD88 signaling pathway, block the OTA-mediated NF-κB signaling pathway, reduce OTA-induced intestinal toxicity, and repair the intestinal barrier [193,194]. Transcriptomics results showed that *Bacillus subtilis* CW14 would activate the toll-like receptor signaling pathway to protect the ZO-1 protein, and minimize OTA-induced cell apoptosis by down-regulating the death receptor gene and up-regulating the DNA repair gene [195].

There is still a dearth of investigation of precise pathogenic pathways related to gut microorganisms of AFB1 and ZEA. As a result, the prevention and treatment of these two mycotoxins are primarily focused on intestinal barrier development and preservation of gut microbial balance. Chen et al., found that *Bacillus amylolitica B10* dramatically increased intestinal tight junction protein expression in mice while decreasing the relative abundance of *Bacteroides* and *Bacteroidetes* [196]. Wang et al. demonstrated that *Bacillus cereus BC7* effectively normalized ZEA-induced disturbances in the intestinal microbiota, and significantly increased *lactobacillus* abundance for microbiome homeostasis in mice, thereby reversing abnormal histological phenotypes in the uterus, ovaries, and liver exposed to ZEA [197].

## 5. Conclusions

Cross-talk between mycotoxins and bacteria is complicated, and requires systematic and extensive studies. Researchers have finally been able to decipher this mysterious black box because of advancements in microbiome culture and sequencing technology. It is through the digestive system that mycotoxins enter the body, and the intestinal microbe is the primary vector for the crosstalk effect that occurs with them. In addition to broadening our horizons in understanding the toxicological mechanisms of mycotoxins, investigating the cross-talk relationship between mycotoxins and microorganisms can also provide us with more valuable and potential therapeutic targets that alleviate the toxic effects of mycotoxins. As the depth and breadth of research increases, more mechanisms to mitigate mycotoxicosis through microorganisms will be revealed and applied to bioengineering production and animal husbandry.

## Figures and Tables

**Table 1 toxins-14-00859-t001:** Direct toxicity of mycotoxins to intestinal epithelial cells.

Mycotoxin Exposure Treatment	Sample Types	Toxicity to Intestinal Epithelial Cells	Reference
DON, 0.5 μM, incubation 6 or 12 h	Pig, intestinal epithelial cells (IPEC-J2)	Activates diamine oxidase (DAO), Significantly decreased expression levels of TFF2, TFF3, and Claudin-4 genes	[24]
DON, 1300 and 2200 μg/kg feed, 60 d	Pig, duodenal epithelial cells	Activates DAO, Low-dose group: endoplasmic reticulum swelling, irregular chromatin distribution; high-dose group: chromatin condensation, nuclear pyknosis, mitochondrial swelling and vacuolization	[25]
DON, 1008 μg/kg feed, 42 d	Pig, Cecal epithelial cells	Decreases numbers of immune cells TLR2 and TLR4 in cecal epithelial cells, up-regulated NFκB signaling pathway	[26]
DON, 0.1, 1, 10, 100 μM, incubation 10–14 d	Human, colon cancer cells (HT29-16E cells) and colorectal adenocarcinoma cells, (CACO-2 cells)	Dose-dependently inhibits the expression of TFF family genes by regulating the expression of protein kinase R and MAP kinase (MAPK) p38 and ERK1/2	[27]
DON, 2 μM, incubation 24 h	Pig, intestinal epithelial cells (IPEC-J2)	Decreases the protein stability and accelerates the degradation of TJP in the lysosome	[28]
ZEA, 6 and 8 μg/mL, incubation 12, 24 and 36 h	Pig, intestinal epithelial cells (IPEC-J2)	Up-regulates ROS, causing oxidative stress	[29]
ZEA, 20 μg/mL, incubation 24 h	Pig, intestinal epithelial cells (IPEC-J2)	Up-regulates ROS, causing oxidative stress	[30]
ZEA, 6 and 8 μg/mL, incubation 24 h	Pig, intestinal epithelial cells (IPEC-J2) and mice, peritoneal macrophages	Increases NLRP3 inflammasome expression and cytokine release	[31]
ZEA, 40 μM, incubation 24 h	Pig, intestinal epithelial cells (IPEC-J2)	Inhibits cell proliferation and causes intestinal cell damage	[32]
OTA, 5, 10, 20, 40, 80 μM, incubation 12 h	Pig, intestinal epithelial cells (IPEC-J2)	Induces ROS generation causes barrier dysfunction, and disrupts tight junctions	[33]
OTA, 0.0005, 0.005 and 4 μg/mL, incubation 48 h	Human, colorectal adenocarcinoma cells, (CACO-2 cells)	Perturbs functional gene expression and induces apoptosis in a dose-dependent manner	[34]
FB1, 10, 25 and 50 μg/mL, incubation 24 or 48 h	Pig, intestinal epithelial cells (IPEC-J2)	Inhibits cell proliferation and damages tight junction proteins	[35]
FB1, 5 mg/kg feed, 42 d	Mice, duodenal epithelial cells	Causes epithelial cells of duodenal villi to slough off and partial necrosis of intestinal glands	[36]
AFB1, 0.6 mg/kg feed, 21 d	Chicken, intestinal epithelial cells	Pathological changes in the ultrastructure of duodenal mitochondria, complete shedding of microvilli on the surface of the jejunum, reduce the number of mitochondria	[37]
AFB1, 0.12 and 12 μM, incubation	Human, colorectal adenocarcinoma cells, (CACO-2 cells)	Disrupts gut-tight junction proteins	[38]

**Table 2 toxins-14-00859-t002:** Effects of mycotoxins exposure on gut microbiota.

Mycotoxin Exposure Treatment	Animal Models	Toxicity to Gut Microbiota	Reference
DON, 10 μg/kg BW, 1–280 d	Mice	Up-regulates the relative abundance of pathogens highly associated with chronic intestinal diseases at the phylum, family, and genus levels.	[39]
DON, 3.02 mg/kg feed, 14–20 d and 35–41 d	Pig	This leads to altered fecal microbiota composition and microbial biological functions associated with mycotoxin detoxification	[40]
DON, 5 mg/kg feed, 1–35 d	Chicken	Promotes the *Campylobacter jejuni* colonization and translocation in intestinal epithelial cells	[41]
DON, 5 mg/kg feed, 1–21 d	Chicken	Manipulates microbiota community composition and metabolite production, disrupts host metabolic processes	[42]
DON, 2.5, 5 and 10 mg/kg feed, 1–35 d	Chicken	Disrupts the composition of the cecal microbiota and reduces the microbial diversity	[43]
DON, 2 and 10 mg/kg BW, 1–28 d	Rat	Exacerbates the genotoxicity of *Escherichia coli*	[44]
DON, 1 and 5 mg/kg, every other day for 1–14 d	Mice	Alters the composition of gut microbiota, affects microbial biosynthetic and degradative functions, and further contributes to host metabolic dysfunction	[45]
OTA, 0.21, 0.5 and 1.5 mg/kg BW, 1–28 d	Mice	Disrupts the structure and diversity of gut microbial communities	[46]
OTA, 235 μg/kg BW, 1–21 d	Duck	Disrupts gut microbial composition and lipopolysaccharide biosynthesis function	[47]
ZEA, 400, 800 and 1600 μg/kg BW, 1–28 d	Rabbit	Affects the cecal microbiota balance and reduces the abundance of bacteria with important metabolic functions	[48]
ZEA, 0.2, 1 and 5 mg/kg BW, 1–28 d	Rat	Disrupts the integrity and function of the mucus layer, induces imbalance of gut microbiota	[49]
ZEA, 5, 10 and 15 μg/kg BW, 1–42 d	Pig	Affects the colony counts of intestinal microbiota	[50]
ZEA, 20 and 40 μg/kg BW, 1–14 d	Mice	Inhibits the glycerophospholipid metabolic pathway of gut microbiota	[51]
ZEA, 20 mg/kg BW, 1–21 d	Mice	Disrupts microbial metabolism of lipid molecules and organic acids	[52]
ZEA, 0.8 mg/kg feed and DON, 8 mg/kg feed, 1–28 d	Pig	The combination of DON and ZEA disrupts the gut microbial composition	[53]
ZEA, 1.36 mg/kg feed and DON, 0.87 g/kg feed, 1–28 d	Pig	Alters gut microbial composition and downregulates abundance of the microbial ribosome and pentose phosphate pathway functional genes	[54]
AFB1, 5, 25, and 75 μg/kg BW, 1–28 d	Rat	Disrupts gut microbiota-dependent organic acid metabolism	[55]

**Table 3 toxins-14-00859-t003:** The degradation rate of mycotoxins by different microorganisms.

Mycotoxin	Microbiota	Culture Conditions	Degradation Rate	Reference
AFB1	*Bacillus shackletonii*	37 °C, 72 h	92.1%	[57]
	*Pseudomonas fluorescens*	37 °C, 72 h	99%	[58]
	*Bacillus sp. Strains*	37 °C, 72 h	58–96.9%	[59]
	*lactobacillus plantarum*	37 °C, 24 h	56%	[60]
OTA	*Brevibacterium*	30 °C, 6 d	100%	[61]
	*bacillus amyloliquefaciens. ASAG1*	31 °C, 72 h	99.7%	[62]
	*Bacillus. CW14*	30 °C, 24 h	97.6%	[63]
FB1	*S. marcescens 329-2*	25 °C, 24 h	37%	[64]
	*L. plantarum MYS6*	30 °C, 30 d	61.7%	[65]
	*lactobacillus plantarum CECT 749*	25 °C, 15 d	90.6%	[66]
	*Lactobacillus brevis*	18–25 °C, 141 d	90%	[67]
ZEA	*E. coli-Lactobacillus shuttle vector pNZ3004*	14 h	99.3%	[68]
	*Lactobacillus reuteri*	37 °C, 4 h	100%	[69]
	*R. percolatus JCM 10087*	28 °C, 7 d	90%	[70]
	*Bacillus subtilis ANSB01G*	37 °C, 48 h	100%	[71]
DON	*Pelagibacterium halotolerans ANSP101*	40 °C, 6 h	81%	[72]

**Table 4 toxins-14-00859-t004:** Compounds that mitigate the effects of mycotoxins.

Mycotoxin Exposure Treatment	Antidote Treatment	Animal Models	Reference
AFB1, 1 mg/kg feed, 1–28 d	Curcumin, 300 mg/kg feed, 1–28 d	Chicken	[84]
AFB1, 0.75 mg/kg BW, 1–30 d	Curcumin, 200 mg/kg BW, 1–30 d	Mice	[85]
AFB1, 0.75 mg/kg BW, 1–21 d	Curcumin, 400 mg/kg feed, 1–21 d	Duck	[86]
AFB1, 0.75 mg/kg BW, at 70 d	Curcumin, 500 mg/kg feed, 1–70 d	Duck	[87]
AFB1, 0.75 mg/kg BW, at 70 d	Curcumin, 500 mg/kg feed, 1–70 d	Duck	[88]
AFB1, 50 μg/kg BW, 1–12 d	Caffeic acid, 40 mg/kg BW, 1–12 d	Rat	[89]
AFB1, 168.3 μg/kg BW, 1–58 d	Myoinositol, 527.9 mg/kg feed, 1–58 d	Litopenaeus vannamei	[90]
AFB1, 1 mg/kg feed,1–28 d	Proanthocyanidin, 250 mg/kg feed, 1–28 d	Chicken	[91]
AFB1, 1 mg/kg feed,1–28 d	Proanthocyanidin, 250 mg/kg feed, 1–28 d	Chicken	[92]
AFB1, 500 μg/kg feed,1–60 d	Silymarin, 500 mg/kg feed, 1–60 d	Japanese quail	[93]
AFB1, 100 µg/kg feed, 1–42 d	Lycopene, 200 mg/kg feed, 1–42 d	Chicken	[94]
AFB1, 0.75 mg/kg BW, 1–31 d	Lycopene, 5 mg/kg BW, 1–31 d	Mice	[95]
AFB1, 0.75 mg/kg BW, 1–31 d	Lycopene, 5 mg/kg BW, 1–31 d	Mice	[96]
AFB1, 0.75 mg/kg BW,1–28 d	Luteolin, 50 mg/kg BW,1–28 d	Mice	[97]
AFB1, 300 μg/kg BW, 1–42 d	Ferulic acid, 120 mg/kg BW,1–42 d	Rat	[98]
AFB1, 100 μg/kg feed, 1–28 d	Marine-algal polysaccharides, 2500 mg/kg feed, 1–28 d	Chicken	[99]
AFB1, 100 mg/kg BW, 1–14 d	Fucoidan, 200 mg/kg BW, 1–14 d	Rat	[100]
AFB1, 0.1 mg/kg BW, 1–28 d	Selenium, 1 mg/kg BW, 1–28 d	Duck	[101]
AFB1, 0.3 mg/kg feed, 1–21 d	Selenium, 0.6 mg/kg feed, 1–21 d	Chicken	[102]
AFB1, 0.3 mg/kg BW, at 70 d	Resveratrol, 500 mg/kg feed, 1–70 d	Duck	[103]
AFB1, 74 μg/kg BW, 1–21 d	Lipoic acid, 300 mg/kg feed, 1–21 d	Chicken	[104]
DON, 4 mg/kg feed, 1–14 d	0.5% Baicalin-Zinc complex feed, 1–14 d	Weaned piglets	[105]
DON, 4 mg/kg feed, 1–21 d	Baicalin-Copper, 5 g /kg feed, 1–21 d	Weaned piglets	[106]
DON, 4 mg/kg feed, 1–14 d	0.1% Baicalin feed, 1–14 d	Weaned piglets	[107]
DON, 3 mg/kg BW, 1–12 d	Lycopene, 10 mg/kg BW, 1–12 d	Mice	[108]
DON, 3.8 mg/kg feed, 1–28 d	Resveratrol, 300 mg/kg feed, 1–28 d	Weaned piglets	[109]
DON, 3.8 mg/kg feed, 1–28 d	Resveratrol, 300 mg/kg feed, 1–28 d	Weaned piglets	[110]
DON, 2.65 mg/kg feed, 1–21 d	Resveratrol, 300 mg/kg, 1–21 d	Weaned piglets	[111]
DON, 4 mg/kg feed, 1–28 d	0.2% Sodium butyrate feed, 1–28 d	Weaned piglets	[112]
DON, 2 mg/kg BW, 4–9 d	Zinc L-Aspartate, 20 mg/kg BW, 1–6 d	Mice	[113]
DON, 3 mg/kg BW, 1–10 d	l-Carnosine, 300 mg kg BW, 1–10 d	Mice	[114]
DON, 3 mg/kg BW, 1–15 d	Ginsenoside Rb1, 50 mg/kg BW, 1–15 d	Mice	[115]
DON, 2 mg/kg BW, 1–10 d	Lauric acid, 10 mg/kg BW, 1–10 d	Mice	[116]
DON, 1 mg/kg BW, 11–17 d	Chloroquine, 10 mg/kg BW, 4–10 d	Weaned piglets	[117]
DON, 2 mg/kg BW, 4–8 d	Methionine, 300 mg /kg BW, 1–11 d	Mice	[118]
DON, 4 mg/kg feed, 1–36 d	2% glutamic acid feed, 1–36 d	Weaned piglets	[119]
FB1, 1 mg/kg BW, 1–28 d	Alginate oligosaccharides, 200 mg/kg BW, 1–28 d	Mice	[120]
FB1, 400 ppb feed, 1–42 d	Glycerol monolaurate, 4 mg/kg feed, 1–42 d	Chicken	[121]
FB1, 5 mg/kg feed, 1–42 d	Moringa leaf, 20 g/kg feed, 1–42 d	Rabbit	[122]
ZEA, 0.25 mg/kg feed, 8–14 d	Betulinic acid, 0.5 mg/kg feed, 1–14 d	Mice	[123]
ZEA, 20 mg/kg feed, 1–42 d	Silymarin, 500 mg/kg feed, 1–42 d	Rat	[124]
ZEA, 0.27 mg/kg feed, 1–14 d	Fructo oligosaccharide10 g/d feed, 1–14 d	Cattle	[125]
ZEA, 40 mg/kg feed, 1–30 d	Hyperoside, 100 mg/kg feed, 1–30 d	Mice	[126]
ZEA, 5 mg/kg feed, 1–7 d	Baicalin, 80 mg/kg feed, 5–7 d	Chicken	[127]
ZEA, 2 mg/kg feed, 1–21 d	Resveratrol, 5 mg/kg feed, 1–21 d	Rat	[128]
ZEA, 40 mg/kg BW, 13–14 d	Selenium yeast, 0.25 g/kg BW 1–14 d	Mice	[129]
ZEA, 40 mg/kg BW, 11 d	Proanthocyanidin, 75 mg/kg BW, 1–10 d	Mice	[130]
ZEA, 40 mg/kg BW, 6–7 d	Proanthocyanidin, 100 mg/kg BW, 1–5 d	Mice	[131]
ZEA, 40 mg/kg BW, 11 d	Lycopene, 20 mg/kg BW, 1–10 d	Mice	[132]
ZEA, 100 µg/kg BW, 1–28 d	Gallic acid, 40 mg/kg BW, 1–28 d	Rat	[133]
ZEA, 0.725 mg/kg feed, 1–14 d	Garlic 30 g/kg and chitosan 10 g/kg feed, 1–14 d	European seabass	[134]
ZEA, 40 mg/kg BW, 1–28 d	Selenium, 0.4 mg/kg BW, 1–28 d	Mice	[135]
ZEA, 40 mg/kg BW, 11 d	Chrysin, 20 mg/kg BW, 1–10 d	Mice	[136]
OTA, 0.5 mg/kg feed, 1–42 d	Quercetin, 0.5 g/kg feed, 1–42 d	Chicken	[137]
OTA, 3 mg/kg BW, 1–21 d	Quercetin, 50 mg/kg BW, 1–21 d	Rat	[138]
OTA, 0.5 mg/kg BW, 1–45 d	Curcumin, 100 mg/kg BW, 1–45 d	Rat	[139]
OTA, 0.5 mg/kg BW, 1–45 d	Curcumin, 100 mg/kg BW, 1–45 d	Rat	[140]
OTA, 2 mg/kg feed, 1–21 d	Curcumin, 400 mg/kg feed, 1–21 d	Duck	[141]
OTA, 0.5 mg/kg BW, 1–14 d	Curcumin, 100 mg/kg BW, 1–14 d	Rat	[142]
OTA, 5 mg/kg BW, 1–21 d	Astaxanthin, 100 mg/kg BW, 1–21 d	Mice	[143]
OTA, 5 mg/kg BW, 1–21 d	Astaxanthin, 100 mg/kg BW, 1–21 d	Mice	[144]
OTA, 50 μg/kg BW, 1–21 d	Selenium yeast, 0.4 mg/kg BW, 1–21 d	Chicken	[145]
OTA, 50 μg/kg BW, 1–21 d	Selenium yeast, 0.4 mg/kg BW, 1–21 d	Chicken	[146]
OTA, 1 mg/kg BW, 1–21 d	Gallic acid, 280 mg/kg BW, 1–21 d	Catfish	[147]

## Data Availability

The data presented in this study are available in this article.

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
