# Peer review of "Crosstalk between Mycotoxins and Intestinal Microbiota and the Alleviation Approach via Microorganisms"

_toxins, 2022, doi:10.3390/toxins14120859_

Round 1

Reviewer 1 Report

The review critically addresses an important and interesting topic in a well written and organized way.

I have minor comments suggestions, as follows:

-Please refer to Table 1 in the text of section 2. and summarise the main findings on the in vitro studies on intestinal toxicity of mycotoxins.

- Please refer to Table 2 in the text of section 3.

- To harmonise the structure between sections 2. and 3., I suggest reorganizing section 3. in subsections, e.g., 3.1 DON, 3.2 ZEA, etc.

 Line 255 – please amend the Table number (Table 3).

Line 504 – I do not understand the beginning of the last paragraph “More prognostic ways of mycotoxin poisoning mediated by microbes…” – what are the prognostic ways that the authors refer to ? I suggest to rephrase this part.

Author Response

Response to Reviewer 1 Comments

Point 1: Please refer to Table 1 in the text of section 2. and summarise the main findings on the in vitro studies on intestinal toxicity of mycotoxins.

Response 1: Thank you for your advice! We have already referred to Table 1 in section 2 (Line 84). Also added about the second part of the summary (Line 176-179).

Point 2: Please refer to Table 2 in the text of section 3.

Response 2: Thanks for your advice! We have already referred to Table 2 in the body of section 3 where appropriate (Line 184).

Point 3: To harmonise the structure between sections 2. and 3., I suggest reorganizing section 3. in subsections, e.g., 3.1 DON, 3.2 ZEA, etc.

Response 3: Thanks for your suggestion! We have added the subheadings of all the paragraphs in section 3.

Point 4: Line 255 – please amend the Table number (Table 3).

Response 4: Thanks for your advice! I'm sorry that it was a clerical error and we have corrected it (Line 290).

Point 5: Line 504 – I do not understand the beginning of the last paragraph “More prognostic ways of mycotoxin poisoning mediated by microbes…” – what are the prognostic ways that the authors refer to ? I suggest to rephrase this part.

Response 5: Thanks for your advice! The ambiguity here has been removed and we have rewritten this part (Line 551-553).

Reviewer 2 Report

Manuscript ID: toxins-1760281

Title: The mediating role of intestinal microorganism in mycotoxin-induced inflammation and their potential value as therapeutic targets

Comments to the Author  

In this article, the authors attempt to review the potential role between five major mycotoxins (AFB1, ZEN, DON, FB1 and OTA) and gut microbes for understanding the toxicological process, prevention and control of mycotoxins. Overall, the manuscript is well written and updated issue. The review has been almost covered the recent scientific-based information. Together with this, this review article also makes a significant point and informative following the title and objective, however there are some points should be more expanded in the manuscript before it can be acceptable for publication. 

-  Comments

n - In 3. Crosstalk between mycotoxins and intestinal microbiota: The information of FB1 should be added in this section to provide the conclusion completely.

- In 4.2. Alleviate mycotoxins harm by microbiota adhesion effect: The information of microbiota adhesion effects of FB1 and OTA should be expanded in the revised manuscript.

 -   Minor point: The abbreviation of “ZEN” should be changed to “ZEA” in the abstract. (L30)

Author Response

Response to Reviewer 2 Comments

Point1: In 3. Crosstalk between mycotoxins and intestinal microbiota: The information of FB1 should be added in this section to provide the conclusion completely.

Response 1: Thanks for your advice! Due to the lack of sufficient experimental evidence on the changes of intestinal microorganisms in animals exposed to FB1, no summary has been made in the third part. We add one sentence to illustrate this (Line 259-261) In the section 4, we added that some compounds may achieve the effect of alleviating FB1 toxicity indirectly through the regulation of intestinal microorganisms.

Point 2: In 4.2. Alleviate mycotoxins harm by microbiota adhesion effect: The information of microbiota adhesion effects of FB1 and OTA should be expanded in the revised manuscript.

Response 2: Thank you for your advice! The adsorption effect of probiotics on FB1 and OTA is indeed a question worth exploring, but according to the results of our literature search, there is no direct experimental evidence that probiotics can effectively adsorb these two mycotoxins, and the current research on direct microbial action on these two mycotoxins is still limited to fermentation degradation.

But we found that yeast selenium was effective in alleviating the hepatotoxicity and nephrotoxicity caused by OTA, and the results of this test are probably a potential effect of the adsorption of yeast on OTA. We have added to the discussion in our manuscript (Line 490-494).

Point 3: Minor point: The abbreviation of “ZEN” should be changed to “ZEA” in the abstract. (L30)

Response 3: Thank you for your advice. We have amended the abbreviation to ensure consistency (Line 50).